# Thermal–Acoustic Interaction Impacts on Crowd Behaviors in an Urban Park

Ye Chen [1,2], Ziyi Chen [1,2], Shumeng Lin [1,2], Xiaoqian Lin [1,2], Shuting Li [3], Taoyu Li [4] and Jianwen Dong [1,2,*]

[1] College of Landscape Architecture and Art, Fujian Agriculture and Forestry University, 15 Shangxiadian Rd., Fuzhou 350000, China; 2191775001@fafu.edu.cn (Y.C.); 3211726008@fafu.edu.cn (Z.C.); 1221775066@fafu.edu.cn (S.L.); 1221775002@fafu.edu.cn (X.L.)

[2] Engineering Research Center for Forest Park of National Forestry and Grassland Administration, 63 Xiyuangong Rd., Fuzhou 350002, China

[3] College of Economics and Management, Fujian Agriculture and Forestry University, 15 Shangxiadian Rd., Fuzhou 350000, China; 000q822019@fafu.edu.cn

[4] Xiamen Tobacco Industry Co., Ltd., 1 Xinyang Rd., Xiamen 361000, China; lty23208@fjtic.cn

* Correspondence: fjdjw@fafu.edn.cn

**Abstract:** As urbanization accelerates, parks, as vital urban public open spaces, and their acoustic and thermal ambience directly impact visitors' comfort and the sustainability of parks. Selecting Xihu Park in Fuzhou, China located in the subtropical region as a typical example, this study utilizes covert observational experiments with different typical sounds (grass cutting, music, and no sound source) across temperature levels to examine the influence of thermal–acoustic interactions on crowd behaviors in the park. The findings are as follows: (1) melodious music can attract more tourists, while strong stimulating grass cutting noises under high temperatures reduce crowd flow. Excluding unpleasant audio sources, park soundscapes across temperatures have a relatively limited influence on attractiveness to people flow. (2) High temperatures diminish tourists' interest in landscape experiences and persons staying, especially when the soundscape quality is poorer. Under non-high temperatures, audio environments have a minor impact on the staying time. (3) The soundscape quality plays a role by affecting people's path choices of approaching or avoiding sound sources, where grass cutting noise has the most negative influence. Music, grass cutting sounds, and natural sounds demonstrate conspicuous differences in their effects under varied temperatures. (4) Comfortable acoustic environments can draw larger crowds and decrease the walking pace. High temperatures make crowds take slower steps. Different sound types have significant influences on crowd movement velocity under three typical temperature levels. This study comprehensively investigates the mechanisms of typical thermal–acoustic environments' impacts on park crowd behaviors, providing important references for optimizing the acoustic and thermal environments of urban parks, while also enriching related research on landscape design and environmental psychology. Future studies can conduct in-depth explorations by creating more abundant thermal–acoustic combinations and probe differences across diverse populations.

**Keywords:** urban park; crowd behavior; temperature; acoustic; Fuzhou city; soundscape

## 1. Introduction

The breakneck urbanization of the 21st century has spawned manifold environmental predicaments, including water scarcity [1,2], waste accumulation [3], soil contamination [4], overpopulation [5], noise pollution [6], and the specter of climate change [7]. Of particular concern is the pernicious impact of outdoor noise on human health [8–11]. Scientific research corroborates that chronic exposure to cacophony can precipitate hearing impairment, disrupted sleep, and heightened risks of cardiovascular and mental illness. Moreover, the direct repercussions of climate change and the emergence of the urban heat island effect

have bred thermal discomfort in urban areas [12–14]. Nevertheless, urban parks have materialized as an indispensable component of the comprehensive urban ecosystem network, proffering a myriad of benefits for city dwellers [15]. The abundance of sprawling greenery within these parks works to bolster physical and mental well-being, effectively mitigating exposure to air and noise pollution, as well as extreme temperatures. Furthermore, they furnish a sanctuary for psychological relaxation and stress relief [16,17].

Furthermore, urban parks are the cornerstone for the existence, function, and growth of cities [18]. Diverse sensory cues continuously shape people's lifestyles. Thus, the sensory design of municipal green spaces is garnering mounting attention. However, the prevailing issue is the disproportionate prioritization of visual appeal over other sensory modalities. This results in the satisfaction of aesthetic preferences at the sacrifice of visitors' holistic sensory and psychological well-being [19–21].

Moreover perceiving various attributes of the external world makes important contributions to humans' complex cognitive processes, as perception is the cornerstone of many psychological processes [22]. All understanding of the world stems from perception. Moreover, multi-sensory integration can garner richer, more comprehensive information. Existing multi-sensory research has focused primarily on audiovisual [23] and visual–olfactory [24] interactions, while research on thermal–acoustic environment interactions has been limited. For instance, Pellerin (2010) believes initial thermal–acoustic conditions influence subjective sound–heat perception. In noisy environments, noise is the decisive factor. Only when ambient temperature diverges markedly from thermal comfort does temperature become decisive. Additionally, under thermal conditions, noise affects thermal comfort—higher noises increase thermal discomfort at elevated ambient temperatures [25,26]. Concurrently, when environmental temperature differs substantially from subjective thermal perception, auditory perception is weakened. These experiments confirm that different thermal–acoustic combinations can impact psychological parameters and subjective evaluations.

Changes in the sensory environment impact people's psychology. As a result, psychological perception shifts will inevitably lead to corresponding behavioral changes. In sensory environment assessments, behavior plays an important role, as surrounding crowd activities and actions are a critical part of the environment [27]. However, urban park designs are far from ideal due to insufficient research on crowd behavior within them. Hence, it is imperative to study crowd behavior in urban parks.

Based on different research objects, behavior is divided into individual and crowd behavior. Individual behavior typically refers to personal attitudes or performances under specific circumstances, which are largely random under environmental influences [28]. Crowd behavior denotes the activities of groups of people in an environment, following certain regularities [29,30]. Hence, scholars often pay more attention to crowd, rather than individual, behavior when studying urban parks [31]. According to characteristics, crowd behavior can be categorized into movement and action. Movement behaviors include passing through, circling around, and staying still, while actions encompass sitting, standing, sightseeing, and strolling [32]. Movement behaviors can better reflect overall crowd trends, while the scope and extent of the influence on action behaviors are relatively smaller. Research shows that sensory environments impact human behavior. For acoustic influences, music sound is a common auditory stimulus. The presence of music guides crowds to exhibit orientation behaviors towards the sound source, slowing down the pace. Additionally, sitting crowd density decreases with an increasing distance from the source [28,33]. Moreover, music can increase the crowd dwelling time in locations, like pavilions, garden paths, and park plazas, with the music genre also potentially affecting results [34,35]. Thermal environments likewise influence action behaviors, with people exhibiting different evaluations and actions under varying park temperatures [36]. Currently, related research on the thermal–acoustic interaction's impact on crowd behavior to improve urban park utilization remains limited. Overall, studies exploring the relationship

between sound and temperature and their influence on crowd behavior to enhance urban park use are scarce.

This study conducted covert behavioral observation experiments in a typical urban park, using low-, medium-, and high-heat temperatures as different temperature levels, while replicating typical urban park music sound sources and lawn mower sounds (as well as no sound conditions). The aim was to examine the regularities of crowd movement behaviors under the interaction of sound and temperature, thereby improving the utilization of urban parks. The research questions addressed in this study are as follows: (1) the impact of thermal–acoustic interactions on the number of people; (2) the impact of thermal–acoustic interactions on the number of persons staying; (3) the impact of thermal–acoustic interactions on path offset; (4) the impact of thermal–acoustic interactions on crowd speed.

## 2. Materials and Methods

### 2.1. Study Site Overview

This research was conducted in Fuzhou, a typical city in the humid subtropical region of China. Fuzhou has a pleasant and humid climate with evergreen seasons, abundant sunshine, and rainfall, as well as a short winter and long summer. The frost-free period reaches up to 326 days annually. The average annual sunshine hours are between 1700 and 1980, and the average annual rainfall is between 900 and 2100 mm. The average annual temperature is 20–25 °C, with the coldest month being December, with an average temperature of 6–10 °C, and the hottest month being July and August, with an average temperature of 33–37 °C. The extreme temperatures range from 2.5 to 42.3 °C annually. The average relative humidity is about 77%, and the city often experiences the urban heat island effect due to its basin topography, where the temperature can reach over 36 °C at noon in the summer. The dominant wind direction is northeast, with south wind prevailing in summer [37]. The research site was selected based on whether it had the common sound sources and thermal environments found in urban parks. To investigate the effect of the thermal acoustic environment on subjective evaluations of urban parks, taking into account factors, such as the park area and its accessibility, the Xihu Park located in the city center was finally chosen.

Xihu Park is located in the northwest part of Gulou District, Fuzhou City, and is situated at the city's center, adjacent to residential, commercial, and office areas in the city center. It is only a 700 m away from the main transportation artery Yangqiao East Road and can be reached by a 10 min walk. With a history of more than 1700 years, it is the most well-preserved classical garden in Fuzhou, and it is renowned as the "Pearl of Fujian Gardens", ranking among the top 36 West Lakes in the country. Currently, it covers an area of 42.51 hectares, with a land area of 12.21 hectares and a water area of 30.3 hectares. The vegetation composition in Xihu Park is complex, dominated by forests, grasslands, shrub beds and flower gardens. The forest area is relatively large, consisting primarily of deciduous and mixed coniferous forests. The deciduous broad-leaved forest in the Huxin Mountain is the most typical, characterized by dominant tree species, such as nanmu, camphor, banyan, and paulownia, forming a natural, primitive and spectacular ecosystem. The grasslands consist of common lawn grass and some wildflowers, while the shrubs are composed of flower bushes and some climbing plants. The rich and diverse vegetation composition of Xihu Park is functionally complete, providing an ideal place for urban ecological construction, as well as for citizens to relax, exercise, and enjoy their leisure time [38].

There were a total of 2 sample sites in Xihu Park. The panoramic view of these sites can be seen in the accompanying Figure 1. On a representative summer day (a weekday in September), crowd behavior observation and thermal acoustic monitoring was conducted.

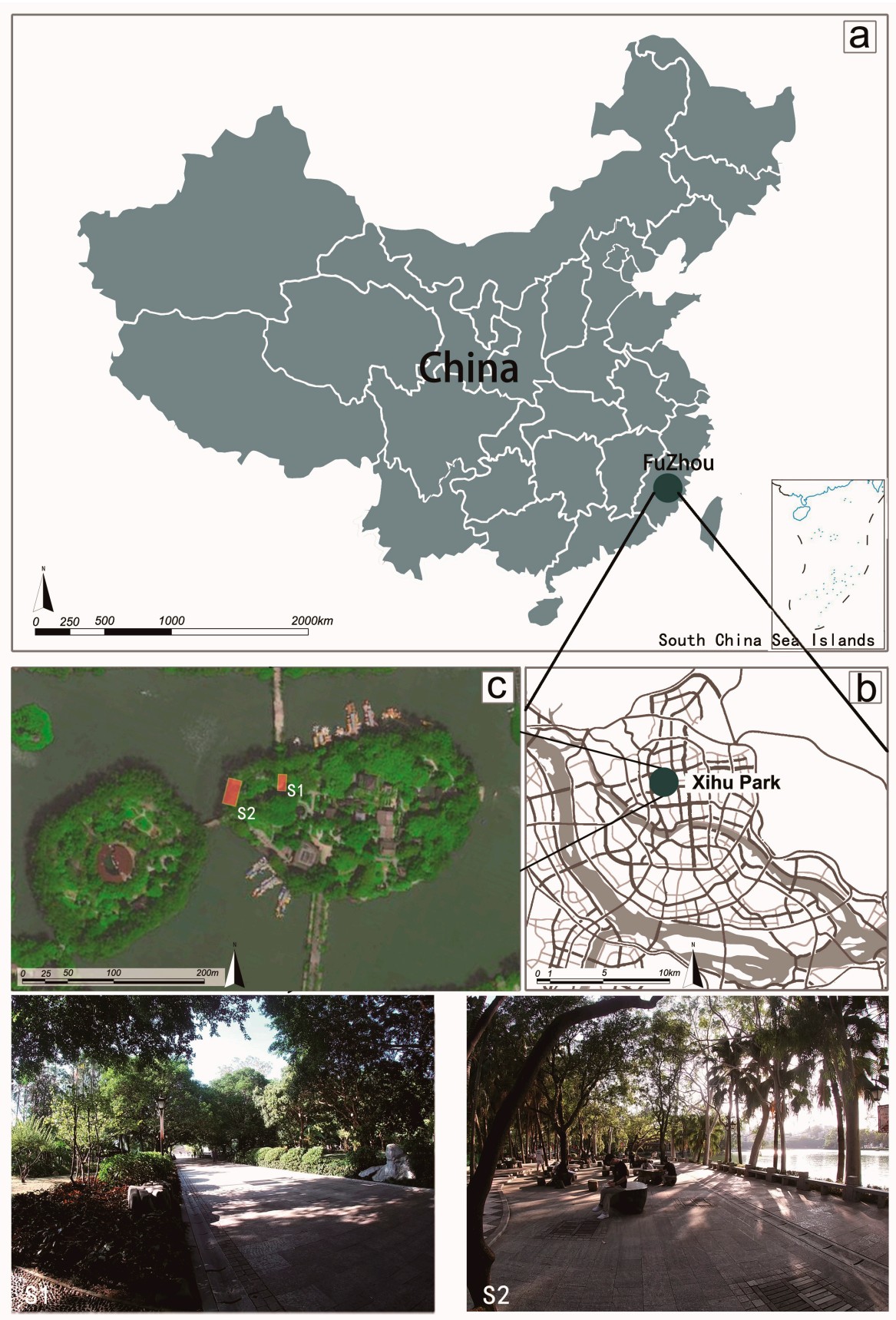

**Figure 1.** (**a**) Location of Fuzhou city in the China map; (**b**) Xihu Park in Fuzhou city; (**c**) S1 and S2 sample sites within Xihu Park.

## 2.2. Procedures

For objective acoustic indicators, this study has selected LAeq (A-weighted equivalent continuous sound level) as the primary monitoring metric owing to the discontinuity of park soundscapes [39]. For thermal environment metrics, the Universal Thermal Climate Index (UTCI) was chosen given the instability of outdoor thermal conditions [40]. UTCI denotes the equivalent temperature of a reference environment eliciting identical physiological responses as a standard individual in the actual environment. This index synthesizes subjective factors and objective environmental parameters, accounting for human thermal adaptability [32,41]. The UTCI value is derived from thermal environmental parameters gathered via the questionnaire. Once data on the radiation temperature, air temperature, relative humidity, wind speed, and mean radiant temperature are prepared, the UTCI can be calculated using either the BioKlima2.6 software or the Fortran program from the official website (http://www.utci.org accessed on 3 August 2023).The deviation of the UTCI from air temperature is determined by the actual air temperature (Ta), mean radiant temperature (Tmrt), wind speed (Va), and relative humidity (Rh). This can be expressed as follows [42]:

$$\text{UTCI} = Ta + \text{Offset}(Ta, Tmrt, Va, Rh) = f(Ta, Tmrt, Va, Rh) \tag{1}$$

where Ta denotes the temperature of a 2 m air column, Va denotes the wind speed at 10 ms, Pa denotes the relative humidity, and Tmrt denotes the average radiative temperature.

## 2.3. Selection of the Thermal–Acoustic Environment

In terms of urban parks, there is a rich variety of sounds, including natural sounds from the plant kingdom, such as birds singing, frogs croaking, insects chirping, and the rustling of leaves in the wind, among others. In addition, due to the extensive activities of people in the park, artificial noise is almost everywhere. Furthermore, maintenance work also generates noise, such as construction sounds, traffic noise, and the sound of machinery operating, among others. Therefore, it is rare for urban parks to have a single sound environment or to be completely silent. In comparison, controlling the thermal environment is relatively easier. However, in practical situations, it is difficult to find natural conditions that can simultaneously maintain sound consistency but vary in thermal conditions or maintain thermal consistency but vary in sound. Therefore, a field site with a controllable thermal environment was selected as the research area, and the combination of sound and thermal factors was controlled by playing sounds.

### 2.3.1. Thermal Environment

Given the differences in lifestyles and urban environments among tourists, there are certain limitations to the soundscape experiment. This study primarily focuses on the conversation between individuals and natural soundscapes in a suitable warm environment, under normal temperature conditions for outdoor activities. For instance, outdoor activities are best suited during summer mornings and evenings. However, during midday, people may react to the intense sunlight, which could shift the focus of the sound environment observation experiment and consequently affect the accuracy of the data. When selecting the experimental site, the following factors should be considered: firstly, the thermal environment of the site should be relatively stable. Secondly, the sound pressure level and type of sound source in the sound environment should remain relatively stable, without any interference from sudden ambient sound sources. The selected observation site in the high-temperature area is located on the park road of Xihu Park in Fuzhou City. This road has minimal obstructions, and higher temperatures can be clearly felt when walking in the high-temperature area. The observation sites in the low- and medium-temperature areas are situated on the recreational plaza of Xihu Park in Fuzhou City. In the low- and medium-temperature environments, visitors have more options for activities, and the plaza provides enough seating to meet their needs.

The chosen observation site in the high-temperature region is located on the Garden Road in Xihu Park, Fuzhou. There are not many obstructions on this road, resulting in higher temperatures. While walking in the high-temperature region, one can clearly feel the heat wave hitting their face. The observation sites in the low-temperature and medium-temperature regions are situated on the leisure plaza in Xihu Park, Fuzhou. In the low-temperature and medium-temperature environments, visitors have a greater variety of activities to choose from, and the plaza provides ample seating to meet their needs. The UTCI difference among the low-temperature, medium-temperature, and high-temperature regions should be at least equal to or greater than 1.

2.3.2. Acoustical Environment

This study considers using artificial voices as sound source variables through speakers, with both positive and negative impacts on humans. Ultimately, the researchers selected the typical sound sources of a lawn mower and music in a city park.

The lawn mower is a necessary tool for maintaining the normal quality of the park's lawn by trimming overgrown grass, making it a characteristic sound source in the park. Prior to the formal experiment, a pre-experiment was conducted to select the grass cutting sound [43]. Various grass cutting sounds from different city park environments were first chosen, followed by randomly selecting 30 participants to evaluate the annoyance level of these grass cutting sounds on a seven-point scale (not annoying 1-2-3-4-5-6-7 very annoying). The grass cutting sound with the highest score exceeding 6 was ultimately chosen as the sound source for this experiment.

Urban parks should provide a relaxing and comfortable atmosphere for people; therefore, when selecting the genre of music, consideration should be given to using light music. Light music refers to instrumental pieces within popular music, characterized by simple structure and beautiful melodies, creating a leisurely ambiance. The tempo of the music is also crucial in the selection process. According to research, a slow tempo is defined as 40–70 bpm, and a moderate tempo falls between 85 and 110 bpm, while anything above 120 bpm is considered fast-paced [44]. Furthermore, studies indicate that music below 80 bpm is associated with negative emotions and can lower the heart rate, whereas music above 120 bpm is linked to positive emotions but can increase the heart rate and breathing speed [45]. Therefore, choosing music with a moderate tempo is more suitable. Following careful research, the Urban Park finally decided to use "Snowdreams" by Bandari, a track that has gained popularity based on online charts, with a tempo of 85 bpm.

Therefore, in this study, sound variables were categorized into three conditions: the natural condition with no playing sounds, the condition of playing lawnmower sounds, and the condition of playing music sounds. When selecting the placement of speakers, several considerations were taken into account: firstly, ensuring that the sound is audible from any position within the area; secondly, maintaining a distance of at least 1.5 m between the front of the speaker and any reflective surfaces [46]; finally, in order to avoid any visual impact, the operators concealed the speakers within the research area [28].

*2.4. Objective Measurement of the Sensory Environment*

The devices used in this study are delineated in Table 1. Before the experiments, the research area was divided into grids to enable subsequent measurements and analyses. The grid dimensions for the low-, medium-, and high-temperature test zones were uniformly established at 3 m × 3 m. To guarantee audibility of the speaker playback, the experimenter calibrated the sound pressure level above the background observed without playback and quantified the levels before and after playback to evaluate the acoustic settings. During the experiment, crowd density, as a dynamic factor, was consistently under 0.05 persons per square meter, allowing its impact on sound pressure and visual perception to be discounted [23,47].

**Table 1.** Measuring instruments and uses.

| Instrument | Example | Model | Use |
|---|---|---|---|
| Sound level meter | | BSWA801 | Measure the sound pressure level |
| Eight-channel high-fidelity recorder | | SQuadriga II BHS I | Recording different kinds of sounds |
| Camera | | GoPro 7 | Recording crowd behavior |
| Handheld Mini Weather Station | | Kestrel5500 | Record air temperature, relative humidity, wind speed and direction |

*2.5. Behavioural Observation*

To preclude the environmental influence, measurements were conducted on weekdays in September. Average temperatures that month ranged from 23 to 30 °C, with a relatively stable humidity and wind velocity and direction. Considering the importance of a sufficient sample size, testing times were chosen between 3 PM and 6 PM daily. To mitigate the observational impact on results, individuals were filmed with concealed cameras where typical acoustic factors were present. Participants were unaware of being observed. Each video lasted 20 min, and to ensure random behavior, three measurement sets were taken during each interval [29]. Both sound and silent conditions cycled throughout the experiment [48]. On testing days, no abrupt noises or temperature fluctuations interfered within the research area.

*2.6. Analysis Method*

The experimental results comprise no missing values, and outliers were eliminated or replaced to ensure data accuracy and reliability. Post data collection and organization, objective parameters underwent standardization for calculating the mean and standard deviation of each variable.

In the production and experiments, outcomes relate to certain factors or result from multiple influences. Spearman correlation analysis is a non-parametric statistical method used primarily to evaluate the correlation between two ordinal or ranked variables [49]. Meanwhile, analysis of variance (ANOVA) is a technique inferring if one or more varied factors significantly impact the results. The observed data were analyzed using SPSS 27.0 software (developed by IBM, Armonk, NY, USA) in this study. To assess relationships between different sounds, temperature levels, and crowd behavior, Spearman analysis computed correlations between indicators. ANOVA examined differences in the thermal acoustic elements' impacts on crowd behavior, with Mauchly's sphericity test applied. When sphericity assumptions were violated ($p < 0.05$), corrections were performed—Greenhouse-Geisser for Epsilon < 0.75, Huynh-Feldt otherwise [50]. Experimental data underwent validation prior to analysis.

## 3. Results

### 3.1. Basic Environmental Conditions

After measurements and calculations, the thermal environments in this study are shown in Figure 2a for the low-heat area, Figure 2b for the moderate-heat area, and Figure 2c for the high-heat area. The observation sites in the low- and moderate-heat areas were measured on-site using instruments, with UTCI values ranging from 28 °C to 30 °C in the low area and 30 °C to 31 °C in the moderate area. The high-heat area was measured on-site using instruments as well, with UTCI values consistently above 32 °C, sufficiently exhibiting the high temperature requirement of the high-heat area.

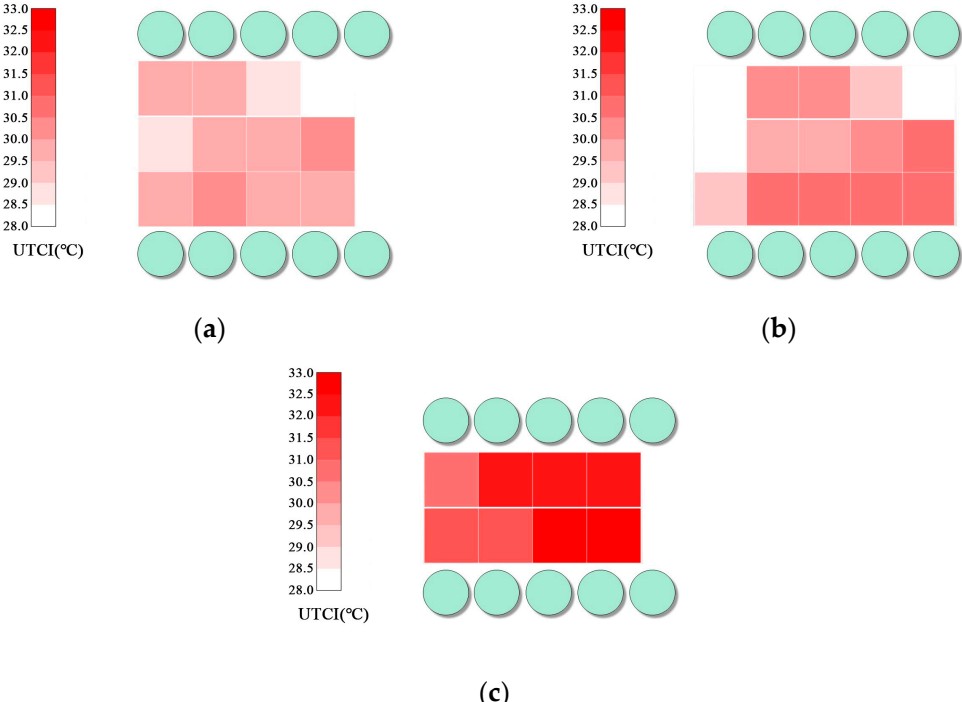

**Figure 2.** Thermal environment situation. (**a**) Low-heat area; (**b**) moderate-heat area; (**c**) high-heat area.

Figure 3a–c display the positions and sound pressure level variations for low- and medium-heat area sound sources. Meanwhile, Figure 3d–f exhibit the positions and variations for high-heat area sources (S denotes the source position). Table 2 provides sample statistics for the observed crowd behavior. During analysis, 50% of samples underwent random retesting, yielding consistent results with the total samples. This indicates sufficiently comprehensive samples and reliable outcomes.

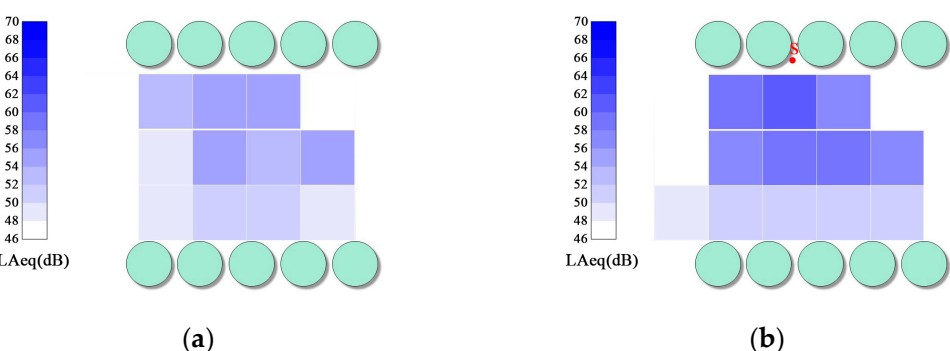

**Figure 3.** *Cont.*

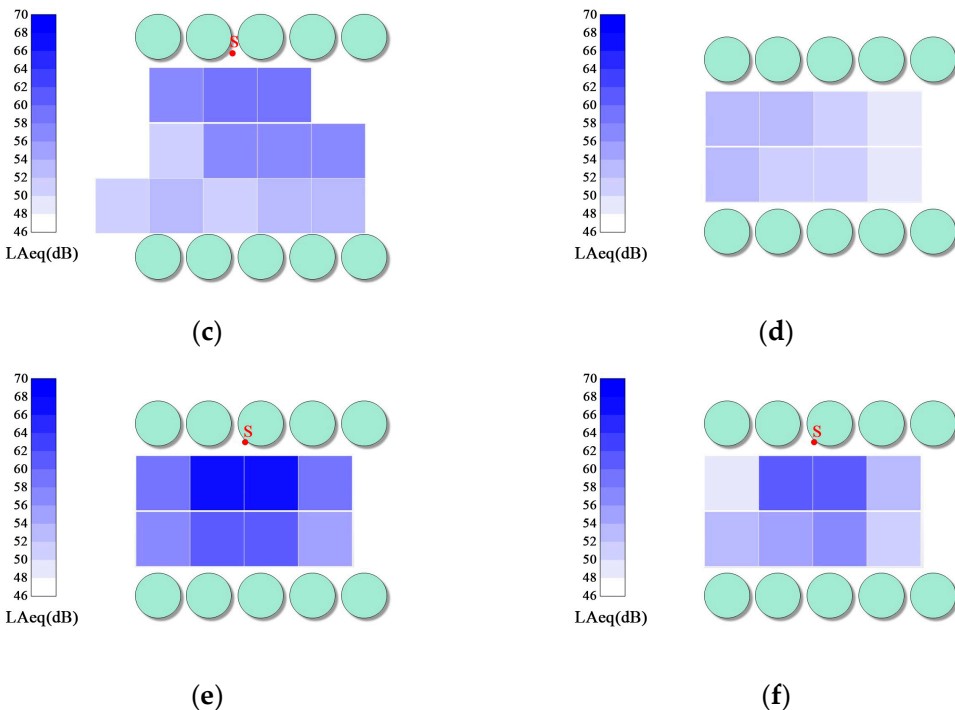

**Figure 3.** Acoustical environment situation: (**a**) without sound played in a low-heat and moderate-heat area; (**b**) with grass cutting sounds in low-heat and moderate-heat areas; (**c**) with music in low-heat and moderate-heat areas; (**d**) without sound played in high-heat area; (**e**) with grass-cutting sounds in high-heat area; (**f**) with music in high-heat area.

**Table 2.** The research samples of behavior observations.

| Temperature Level | Sound Type | Number of People | Number of Persons Staying | Path Offset | Crowd Speed |
|---|---|---|---|---|---|
| Low heat | Grass cutting | 97 | 4 | 0.128 | 1.348 |
| | The natural condition | 130 | 6 | 0 | 1.301 |
| | Music | 153 | 8 | −0.015 | 1.243 |
| Moderate heat | Grass cutting | 86 | 0 | 0.179 | 1.57 |
| | The natural condition | 143 | 5 | 0 | 1.39 |
| | Music | 102 | 10 | −0.196 | 1.378 |
| High heat | Grass cutting | 79 | 0 | 0.349 | 1.105 |
| | The natural condition | 97 | 0 | 0.02 | 1.031 |
| | Music | 102 | 0 | −0.118 | 1.077 |

*3.2. Effects of the Thermal–Acoustic Interaction on the Number of People*

3.2.1. The Number of People under the Thermal–Acoustic Interaction

Figure 4 shows the number of people under thermal–acoustic interactions. It can be observed that the number of people under the grass-cutting sound condition was consistently lower than that under the no sound and music conditions, decreasing slightly as the temperature rises. More specifically, under the grass-cutting sound condition, there were 97 people at low heat, 86 at moderate heat, and 79 at high heat. Overall, the grass-cutting sound attracted significantly fewer individuals than the other two sounds across different temperature conditions. In particular, the no-sound condition persistently involved the largest number of people, peaking under the moderate-heat condition. Namely, under the natural-sound condition there were 130 people at low heat, 143 at moderate heat, and 97 at high heat. In contrast, it can be observed that the number of people under the

music sound condition fluctuated under different temperature conditions. Specifically, the number of people was highest under the music sound condition at low heat, reaching 153; under the music sound condition at moderate heat, the number was 102, whereas at high heat, the number under the music sound condition was very close to the number under the no sound condition, both at 102 people.

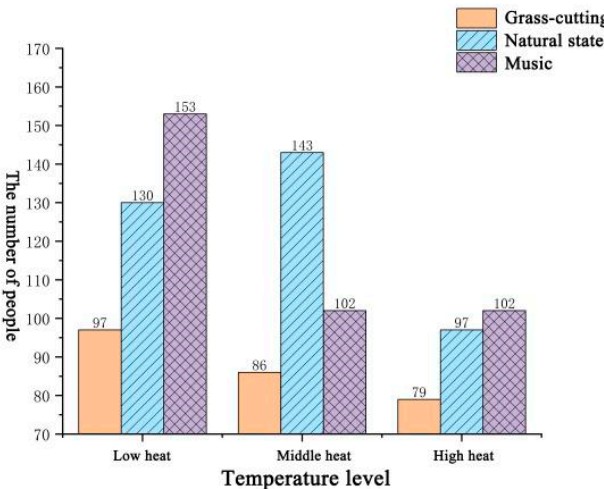

**Figure 4.** The number of people under the thermal–acoustic interaction.

### 3.2.2. Analysis of the Number of People under the Thermal–Acoustic Interaction

The analysis showed a correlation coefficient of 0.718 ($p$ = 0.03 < 0.05) between sound types and the number of people (Table 3), indicating that good acoustics attract more people in urban parks. The coefficient between temperature and the number of people under grass cutting was −1 ($p$ = 0.00 < 0.01) (Table 4), implying that high temperatures reduce crowds with stimulating sounds, like grass cutting. The temperature–crowd correlation was insignificant ($p$ = 0.166 > 0.1), demonstrating that park acoustics do not markedly impact crowd attraction across temperatures without adverse noises. Prior studies show that people gravitate toward pleasing soundscapes. This further elucidates that good acoustics universally attract humans, whether the stimulus is single or multi-sensory.

**Table 3.** Correlation between the thermal–acoustic environment and the number of people.

|  |  | Sound Type | Temperature Level |
|---|---|---|---|
| Number of people | Correlation coefficient | 0.718 * | −0.505 |
|  | Sig. (two-tailed) | 0.030 | 0.166 |

Note: * denotes $p$ (two-tailed) < 0.05. Sig. means significance.

**Table 4.** Correlation between the number of people and temperature under different sound types.

|  |  | Number of People under Grass-Cutting Condition | Number of People under Natural Condition | Number of People under Music Condition |
|---|---|---|---|---|
| Temperature level | Correlation coefficient | −1.000 ** | −0.500 | −0.866 |
|  | Sig. (two-tailed) | 0.000 | 0.667 | 0.333 |

Note: ** denotes $p$ (two-tailed) < 0.01. Sig. means significance.

### 3.3. Effects of the Thermal–Acoustic Interaction on the Number of Persons Staying

3.3.1. The Number of Persons Staying under the Thermal–Acoustic Interaction

Figure 5 shows the number of persons staying under thermal–acoustic interactions. Under the grass-cutting sound condition, the number of persons staying was generally fewer than that under the no sound and music conditions and significantly decreased as

the temperature rose. Specifically, under the grass-cutting sound condition, there were 4 instances of persons staying under low heat, while under moderate- and high-heat, there were 0 instances. Overall, the number of persons staying under the grass-cutting sound was markedly lower than that under the other two sounds across different temperatures. Under the no-added-sound condition, the number of persons staying was highest under low heat. Namely, under natural conditions there were six instances of persons staying under low heat, five instances under moderate heat, and no one staying under high heat. The number of persons staying under the music-sound condition varied under different temperatures, with 8 instances under low heat, 10 instances under moderate heat, and also 0 instances under high heat.

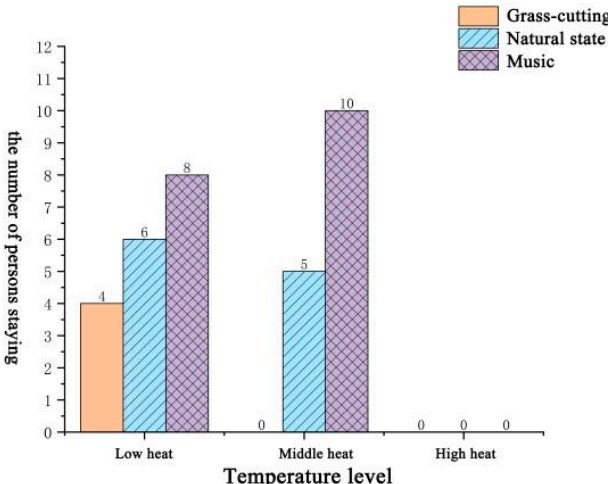

**Figure 5.** The number of persons staying under the thermal–acoustic interaction.

Additionally, under low-heat conditions, there were four instances of persons staying under the grass-cutting sound, six instances under the natural condition, and eight instances under the music sound. Therefore, under low temperature, the music sound attracted more people compared to the other two conditions. Under moderate heat, there were 0 instances under the grass-cutting sound, 5 instances under the natural condition, and 10 instances under the music sound. Hence, under medium temperature, the music sound had the best attraction effect, followed by the natural environment, while the grass-cutting sound had no attraction in this case. Under high heat, the number of persons staying was 0 under all three sound conditions. In summary, under high temperatures, no sound could effectively attract people to stay in the area regardless of audio interference.

3.3.2. Analysis of the Number of Persons Staying under the Thermal–Acoustic Interaction

According to Table 5, the correlation coefficient between temperature levels and the number of people staying was $-0.688$ ($p = 0.000 < 0.05$), meaning that in the urban park environment, the higher the temperature, the lower the tourists' willingness to stay and appreciate the scenery. In addition, Table 6 shows that the correlation coefficient between temperature levels and the number of persons staying under natural conditions was $-1$ ($p = 0.000 < 0.01$), implying that under high-temperature conditions, if there is no good sound environment stimulation, the number of persons staying will decrease. However, the correlation between sound types and the number of persons staying was not significant ($p = 0.149 > 0.1$), indicating that regardless of the sound type, as long as the temperature is not too high, the sound environment of urban parks does not have a significant impact on the attractiveness of the number of persons staying. Previous studies have shown that people tend to go to comfortable thermal environments, and this study further demonstrates that whether the stimulation is from a single-sense or multi-sensory interaction, a good thermal environment has universal appeal to human behavior.

**Table 5.** Correlation between the thermal–acoustic environment and the number of persons staying.

| | | Sound Type | Temperature Level |
|---|---|---|---|
| Number of persons staying | Correlation coefficient | 0.523 | −0.688 * |
| | Sig. (two-tailed) | 0.149 | 0.040 |

Note: * denotes *p* (two-tailed) < 0.05. Sig. means significance.

**Table 6.** Correlation between the number of persons staying and temperature under different sound types.

| | | Number of Persons Staying under Grass-Cutting Condition | Number of Persons Staying under Natural Condition | Number of Persons Staying under Music Condition |
|---|---|---|---|---|
| Temperature level | Correlation coefficient | −0.866 | −1.000 ** | −0.500 |
| | Sig. (two-tailed) | 0.333 | 0.000 | 0.667 |

Note: ** denotes *p* (two-tailed) < 0.01. Sig. means significance.

### 3.4. Effects of the Thermal–Acoustic Interaction on the Path Offset

3.4.1. The Path Offset under the Thermal–Acoustic Interaction

Figure 6 shows the path offset under the interaction of temperature and sound. When the grass-cutting sound was present, the mean path offsets were all positive values, and as the temperature rose, it resulted in a gradual upward trend in the path offset. Under moderate heat, it increased by 0.051 compared to under low heat and by 0.170 under high heat compared to under moderate heat. When no specific sound source was played, the path offsets under different temperature levels were all very small, basically maintained around 0. The only exception was when the temperature rose to 32 °C, the path offset became 0.02, but it was still a negligible increase. When light music was present, the mean path offsets were all negative values. Under low-heat conditions, the path offset was −0.015; under moderate heat, the path offset was −0.196, whereas under high heat, the path offset was −0.118.

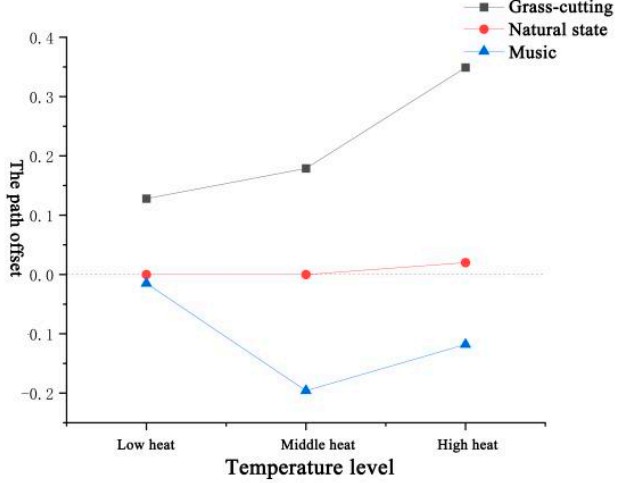

**Figure 6.** Path offset under the thermal–acoustic interaction.

In comparison, under low heat, the path offset under the grass-cutting sound was the largest, followed by that under music sound, while the path offset under natural conditions was the smallest at zero. In comparison, under moderate heat, the path offset under natural conditions remained 0, while the impacts of grass cutting and music sounds were more prominent in comparison. Under high heat, the path offset under the grass-cutting sound was the maximum at 0.349, higher than that under natural and music conditions.

3.4.2. Analysis of the Path Offset under the Thermal–Acoustic Interaction

The analysis results show that the correlation coefficient between sound types and path offset was $-0.336$ ($p = 0.000 < 0.01$), as shown in Table 7. This means that in the urban park environment, good soundscapes can attract tourists to approach the sound source, while undesirable sounds make visitors feel uneasy and stay away.

**Table 7.** Correlation between the thermal–acoustic environment and the path offset.

|  |  | Sound Type | Temperature Level |
|---|---|---|---|
| Path offset | Correlation coefficient | −0.336 ** | 0.072 |
|  | Sig. (two-tailed) | 0.000 | 0.112 |

Note: ** denotes $p$ (two-tailed) < 0.01. Sig. means significance.

The results of multivariate analysis of variance are shown in Table 8. The results indicate that both sound types ($p = 0.000$) and temperature ($p = 0.092$) have significant self-effects on the path offset ($p < 0.1$). As for the interaction between sound and temperature factors, the interaction between sound types and temperature levels ($p = 0.014$) significantly affected the path offset ($p < 0.05$). Therefore, sound and temperature factors both influence the path offset.

**Table 8.** ANOVAs of path offset.

| Source | III Sum of Squares | df | Mean Square | F | Sig. |
|---|---|---|---|---|---|
| Sound | 8.433 | 2 | 4.216 | 32.712 | 0.000 *** |
| Temperature | 0.618 | 2 | 0.309 | 2.396 | 0.092 * |
| Sound × Temperature | 1.631 | 4 | 0.408 | 3.164 | 0.014 ** |

Note: ***, **, and * denote significance at the 1%, 5%, and 10% level, respectively. Sig. means significance.

The multiple comparisons of mean path offsets under the interaction of sound and temperature are shown in Table 9. For grass-cutting sounds, temperatures are divided into two homogeneous subsets, with low and high heat in one subset, showing significant differences from medium and high heat situations. The path offset was maximum under high-heat conditions. For natural conditions and music playback, there were no significant differences across different temperatures.

**Table 9.** Multiple comparisons of path offset under the influence of the sound type and temperature type.

| Sound Type | Temperature Level | Subset at Alpha = 0.05 | |
|---|---|---|---|
|  |  | 1 | 2 |
| Grass cutting | Low-heat | 0.13 |  |
|  | Moderate-heat | 0.18 | 0.18 |
|  | High-heat |  | 0.35 |
|  | Sig. | 0.810 | 0.099 |
| Natural condition | Low-heat | 0.00 |  |
|  | Moderate-heat | 0.00 |  |
|  | High-heat | 0.02 |  |
|  | Sig. | 0.323 |  |
| Music | Moderate-heat | −0.20 |  |
|  | High-heat | −0.12 |  |
|  | Low-heat | −0.02 |  |
|  | Sig. | 0.126 |  |

Note: Sig. means significance.

The multiple comparisons of mean path offsets under the interaction of temperature and sound are displayed in Table 10. For low-heat conditions, the three sound types were

separated into two homogeneous subsets, where music and natural conditions showed no significant difference, but both differed significantly from grass-cutting conditions, which had the maximum path offset. For moderate heat, there were two homogeneous subsets, with natural conditions and grass cutting in one subset, differing significantly from music conditions, which attracted visitors closer to the source. For high heat, there were also two homogeneous subsets, with music and natural conditions in one subset, differing markedly from grass-cutting conditions, where path offset was most noticeable.

**Table 10.** Multiple comparisons of the path offset under the influence of the sound type and temperature level.

| Temperature Level | Sound Type | Subset at Alpha = 0.05 | |
|---|---|---|---|
| | | 1 | 2 |
| Low-heat | Music | −0.02 | |
| | Natural condition | 0.00 | |
| | Grass cutting | | 0.13 |
| | Sig. | 0.895 | 1.000 |
| Moderate-heat | Music | −0.20 | |
| | Natural condition | | 0.00 |
| | Grass cutting | | 0.18 |
| | Sig. | 1.000 | 0.052 |
| High-heat | Music | −0.12 | |
| | Natural condition | 0.02 | |
| | Grass cutting | | 0.35 |
| | Sig. | 0.312 | 1.000 |

Note: Sig. means significance.

### 3.5. Effects of the Thermal–Acoustic Interaction on Crowd Speed

3.5.1. Crowd Speed under the Thermal–Acoustic Interaction

Figure 7 shows the crowd speed under the interaction of temperature and sound. Whether or not special sounds were played, the crowd speed under low-heat conditions was lower than under moderate-heat conditions. Under moderate heat, the crowd speed was highest under the grass-cutting sound condition. While under high heat, the crowd speed decreased under the grass-cutting condition.

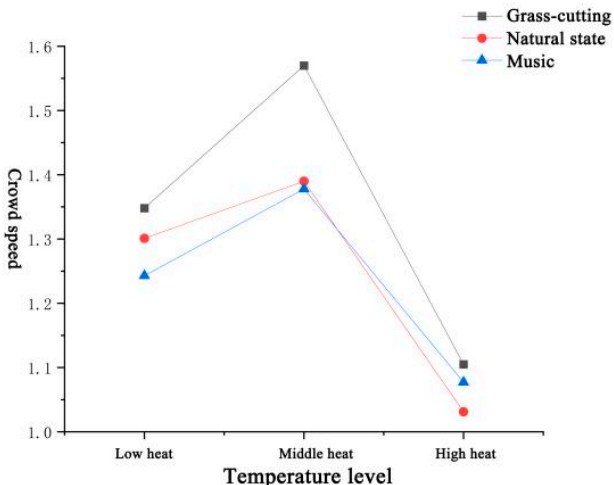

**Figure 7.** Crowd speed under the thermal–acoustic interaction.

Additionally, under low heat, the crowd speed was highest under the grass-cutting condition at 1.348, higher than that under the natural and music conditions. The crowd speed under the natural condition was slightly lower than under the grass-cutting condition at 1.301. The crowd speed was lowest under the music condition at 1.243, somewhat lower

than that under the grass-cutting condition. In comparison, under moderate heat, the crowd speed remained highest under the grass-cutting condition at 1.57, higher than that under both the natural and music conditions. The crowd speed under the natural condition ranked second among the three sound conditions at 1.39. The music condition had the lowest crowd speed at 1.378. Under high heat, the crowd speed was highest under the natural condition at 1.105. The rates under the grass-cutting and music conditions were 1.031 and 1.077, respectively, slightly lower than under the natural condition. However, it should be noted that the crowd speed values decreased under all sound conditions compared to that under the other two temperature conditions.

3.5.2. Analysis of Crowd Speed under the Thermal–Acoustic Interaction

The analysis results show that the correlation coefficient between sound types and crowd speed was $-0.177$ ($p = 0.000 < 0.01$), and the correlation coefficient between temperature levels and crowd speed was $-0.367$ ($p = 0.000 < 0.01$), as shown in Table 11. This means that in the urban park environment, pleasant soundscapes can attract tourists to slow down their pace to better appreciate the park experience. Meanwhile, the higher the temperature, the slower the rate. The possible reason is that people do not want to accelerate their movements, which would make them feel hotter. Studies have shown that air temperature and humidity in high-temperature areas negatively impact human physical activity, affecting people's walking speed. In high-temperature areas, people need to consume more energy to maintain body temperature and fluid balance, leading to a reduced walking rate. Additionally, high temperatures can cause fatigue and discomfort, further slowing down the walking rate.

**Table 11.** Correlation between the thermal–acoustic environment and the crowd speed.

|  |  | Sound Type | Temperature Level |
|---|---|---|---|
| Speed | Correlation coefficient | $-0.177$ ** | $-0.367$ ** |
|  | Sig. (two-tailed) | 0.000 | 0.000 |

Note: ** denotes $p$ (two-tailed) $< 0.01$. Sig. means significance.

The results of multivariate analysis of variance are presented in Table 12. The results indicate that both sound types ($p = 0.000$) and temperature ($p = 0.000$) have significant self-effects on the crowd speed ($p < 0.01$). As for the interaction between the temperature and sound factors, the interaction between sound types and temperature levels ($p = 0.006$) significantly affects crowd speed ($p < 0.01$). Therefore, sound and temperature factors both influence the crowd speed.

**Table 12.** ANOVAs of the crowd speed.

| Source | III Sum of Squares | df | Mean Square | F | Sig. |
|---|---|---|---|---|---|
| Sound | 1.104 | 2 | 0.552 | 15.475 | 0.000 *** |
| Temperature | 10.920 | 2 | 5.460 | 153.035 | 0.000 *** |
| Sound × Temp | 0.517 | 4 | 0.129 | 3.620 | 0.006 *** |

Note: *** denote significance at the 1% level. Sig. means significance.

The multiple comparisons of mean crowd speeds under the interaction of sound and temperature are shown in Table 13. For all sound conditions, the three temperature levels were separated into three homogeneous subsets, with significant differences. It was also observed that medium heat was consistently faster than low and high heat conditions.

The multiple comparisons of mean crowd speeds under the interaction of temperature and sound are displayed in Table 14. For low- and moderate-heat conditions, the three sound types were divided into two homogeneous subsets, where music and natural conditions showed no significant difference, but both differed markedly from grass-cutting

conditions, which had the maximum crowd speed. For high heat, there were no significant differences between sound types.

**Table 13.** Multiple comparisons of the crowd speed under the influence of the sound type and temperature level.

| Sound Type | Temperature Level | Subset at Alpha = 0.05 | | |
|---|---|---|---|---|
| | | 1 | 2 | 3 |
| Grass cutting | High-heat | 1.11 | | |
| | Low-heat | | 1.35 | |
| | Moderate-heat | | | 1.57 |
| | Sig. | 1.000 | 1.000 | 1.000 |
| Natural condition | High-heat | 1.03 | | |
| | Low-heat | | 1.30 | |
| | Moderate-heat | | | 1.39 |
| | Sig. | 1.000 | 1.000 | 1.000 |
| Music | High-heat | 1.08 | | |
| | Low-heat | | 1.24 | |
| | Moderate-heat | | | 1.38 |
| | Sig. | 1.000 | 1.000 | 1.000 |

Note: Sig. means significance.

**Table 14.** Multiple comparisons of the crowd speed under the influence of temperature type and sound type.

| Temperature Level | Sound Type | Subset at Alpha = 0.05 | |
|---|---|---|---|
| | | 1 | 2 |
| Low-heat | Music | 1.24 | |
| | Natural condition | 1.30 | 1.30 |
| | Grass cutting | | 1.35 |
| | Sig. | 0.148 | 0.287 |
| Moderate-heat | Music | 1.38 | |
| | Natural condition | 1.39 | |
| | Grass cutting | | 1.57 |
| | Sig. | 0.959 | 1.000 |
| High-heat | Natural condition | 1.03 | |
| | Music | 1.08 | |
| | Grass cutting | 1.11 | |
| | Sig. | 0.053 | |

Note: Sig. means significance.

## 4. Discussion

### 4.1. Interactions of the Thermal and Acoustic Environment on Crowd Behaviours

Visitors have various purposes in different urban park spaces, which in turn influence crowd behavior differently. This study demonstrates that the thermal–acoustic interaction has a significant impact on crowd behavior, with interactions between sound and varied temperatures of high, medium, and low heat leading to changes in crowd behavior. Therefore, in urban parks, rationally combining specific sounds and temperatures can not only satisfy the usage needs of different functional zones, but also better control visitor flow.

Some urban park spaces require attraction, while others need to guide crowds along proper routes. Research finds that good acoustics in city parks attract crowds. By introducing pleasant natural music, the auditory ambiance of the park can be enhanced. For example, concealed speakers can be installed in the park to play beautiful music or natural sounds, enriching the auditory experience. Meanwhile, the study also indicates that when high temperatures are combined with irritating noise, it can cause discomfort among tourists and lead to their departure. Therefore, the park should suspend noisy operations and provide a quiet space during hot weather. For venues with negative sounds

or thermal factors, like plazas near busy roads, adverse sensory sources should be prioritized for improvements. If the governance effect is limited, positive sounds or temperature regulation should be actively introduced. Previous studies show that people gravitate toward pleasing acoustics [20], further demonstrating the universal appeal of favorable soundscapes regardless of singular or multi-sensory stimuli [51].

Regarding the number of people staying, the study points out that in urban parks, higher temperatures correlate with lower willingness to stay and appreciate the scenery. Without acoustics delivering quality stimulation, instances of staying decrease under high heat. On sound types, park acoustics do not markedly impact attraction, given reasonable temperature ranges. The positive overlapping of sound and temperature can be utilized to increase the willingness to stay, effectively promoting interactions. Adverse sound and thermal factors require appropriate mitigation. For example, to enhance the microclimate of the park, increasing tree shade coverage or providing sunshades can help to alleviate the issue of reduced visiting times for tourists at high temperatures. Past research shows the people favor comfortable thermal settings [52], additionally elucidating the universal appeal of pleasant thermal conditions regardless of single or multi-sensory stimuli [53].

In terms of path offset, the study indicates that in urban parks, pleasant acoustics attract visitors towards sound sources, while unpleasant sounds make people uneasy and repel them. Moreover, sound–temperature interplay also impacts the path offset. Significant differences exist between low and high heat for noises, like grass cutting. Under low and high heat, sound types also differ significantly. Under medium heat, light music markedly differs from other sounds. Therefore, appropriate auditory cues and gradually intensifying or descending temperatures can influence behavior, guiding people along planned routes and avoiding congestion.

Regarding crowd speed, sound and temperature interact to influence velocity based on the results. In urban parks, appealing soundscapes can draw and slow tourists to better enjoy the experience. Meanwhile, higher temperatures are associated with slower paces, likely because people avoid accelerating and overheating [54]. All sound types significantly affect the speed across the three temperature levels. Medium heat is consistently faster than low or high heat. Music and nature do not significantly differ under low or medium heat but substantially differ from grass cutting, where speed is fastest. Under high heat, sound types show no significant differences. By slowing their speed, people can be guided to focus more on the surroundings. Hence, in leisure venues, like urban parks, reducing the crowd speed can enhance the environmental experience [55]. In parks, music or air temperature regulation on paths and plazas can achieve reduced rates when desired.

The findings have an impact on policies and plans aimed at promoting the use of city parks, which plays a crucial role in public well-being. For instance, when there is a need for managing visitor flow in park design and planning, the introduction of specific sounds or adjustments in temperature can influence behavioral tendencies, with certain combinations of sound and temperature reinforcing these effects. Furthermore, different auditory or thermal stimuli in the park environment can have unique effects on crowd behavior. By considering these sensory factors, it becomes possible to accurately predict behaviors in both existing and planned park spaces. Moreover, the use of tailored sounds or the regulation of temperature can mitigate the negative impact of unwanted ambient noises or extreme temperatures on crowd behavior. This analytical approach can be applied to study combinations of different sounds and temperatures, such as addressing noise pollution or tackling excessive heat, to inform decisions on functional zoning, greenery utilization, and government actions related to urban parks.

### 4.2. Shortcomings and Prospects

This study can be expanded in several directions. The experiment was conducted in the afternoon, and crowd physical conditions and psychological perceptions may differ at various times of day. Moreover, the participants' ages may have also influenced the current results; therefore, similar studies should be carried out at other times during the day in the

future. Additionally, future studies should consider other environmental factors related to bodily perceptions that may impact experimental results.

**5. Conclusions**

This study examines the thermal–acoustic interaction's impact on crowd behavior via covert observations combining archetypal thermal–acoustic factors in urban parks. The principal findings are as follows:

(1) In terms of the number of people, favorable acoustics can attract larger crowds. Stimulating sounds, like grass cutting, reduce the crowd size under high heat. Moreover, excluding adverse noises, urban park soundscapes do not significantly impact visitor attraction across temperatures.

(2) Regarding the number of persons staying, high temperatures reduce sightseeing interest, especially with a suboptimal soundscape, prompting departure. Additionally, apart from hot weather, park acoustics do not markedly influence the stay duration.

(3) With respect to the path offset, the acoustic quality determines visitor proximity to sound sources as shown by the path offset, while significant differences emerged in the path offset for grass cutting at varying temperatures, most prominently under high heat. In contrast, natural sounds and music playback do not significantly impact the path offset across temperatures. Under low heat, music and nature do not differ, but both substantially differ from grass cutting, causing the greatest offset. Under moderate heat, nature and grass cutting are analogous, while music attracts visitors. However, under high heat, music and nature are similar but markedly differ from grass cutting, greatly affecting the path offset.

(4) In terms of crowd speed, appealing soundscapes attract and slow tourists to enjoy the park. Moreover, higher temperatures are associated with slower walking. All sound types significantly affect the speed across the three temperature levels. While in moderate heat, speed is consistently faster than in low or high heat. In contrast, in low and moderate heat, music and nature do not significantly differ but substantially differ from grass cutting, where speed is fastest. However in high heat, sound types do not significantly differ.

**Author Contributions:** Conceptualization, Y.C. and J.D.; methodology, Y.C., J.D. and Z.C.; software, Y.C., S.L., T.L. and Z.C.; validation, Y.C., Z.C., S.L. (Shumeng Lin), X.L., T.L. and S.L. (Shuting Li); formal analysis, Y.C., Z.C., S.L. (Shumeng Lin), X.L., T.L. and S.L. (Shuting Li); investigation, Y.C., Z.C., S.L. (Shumeng Lin), X.L., T.L. and S.L. (Shuting Li); resources, Y.C., S.L. (Shumeng Lin), X.L. and T.L.; data curation, Y.C., X.L., T.L. and S.L. (Shuting Li); writing—original draft preparation, Y.C., Z.C., S.L. (Shumeng Lin), X.L., T.L. and S.L. (Shuting Li); writing—review and editing, Y.C. and S.L. (Shuting Li). All authors have read and agreed to the published version of the manuscript.

**Funding:** This research was funded by the (1) Special Project of Wuyishan National Park Research Institute, grant number KJG20009A; (2) Forest Park Engineering Technology Research Center of State Forestry Administration, grant number PTJH15002; (3) Green Urbanization across China and Europe: Collaborative Research on Key technological Advances in Urban Forests, grant number 2021YFE0193200; (4) Horizon 2020 strategic plan: CLEARING HOUSE-Collaborative Learning in Research, Informationsharing, and Governance on How Urban tree-based solutions support Sino-European urban futures, grant number 821242.

**Data Availability Statement:** The data used to support the findings of this study are available from the corresponding author upon request.

**Conflicts of Interest:** The authors declare no conflict of interest.

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
