# Peer review of "Thermal–Acoustic Interaction Impacts on Crowd Behaviors in an Urban Park"

_forests, doi:10.3390/f14091758_

Round 1
Reviewer 1 Report
The research topic is innovative and has practical value. It examines the impact of Thermal-Acoustic interaction on urban park pedestrian behavior, providing reference for optimizing park space. The research scenario is representative, selecting a typical urban park in the subtropical region, and the results can be widely applied.
The research method is systematic and rigorous. The concealed observation experimental method is used, controlling for typical sound sources and temperature levels. The research results and conclusions are highly persuasive.
Overall, this is a high-quality research abstract with a novel topic, rich content, and complete structure. It has both theoretical value and practical significance. The author systematically carried out research design, implementation, and result analysis, achieving significant research outcomes. This paper can be published after the following modifications.
1. The title needs improvement, replacing "interactive influence" with "cumulative influence" may be more appropriate.
2. The photo of the study area in Figure 1 is blurry and has low legibility. It is advised to increase the resolution.
No comments.
Reviewer 2 Report
Comments for Author,
Thank you for your submission. Actually, your paper has a novel content and well documented. In addition to this,
Specific comments:
1-Please rethink the title of paper and use more striking title.
1a.The absract part needs to be expanded a little more.
1b. Please add more properly reference to improve novelty of paper.
2- In my opinion, the original value of the work can be emphasized. Because, although the work is very original, the connections between sentences can be stronger.
3. A more comprehensive recommendation can be made regarding the cocnlusion of the study.
Best Regards
Reviewer 3 Report
The topic is significant for urban planners and event organizers in considering forthcoming rapid climate change. The article is well structured and provides complete views of the conducted study. The methodological part is explained at the detail level, and figures of measurement tools help understand the field research process.
The main question of the research focuses on air temperature impact on visitation length and type of activities in parks in urban areas, particularly in subtropical regions. The research theme is relevant for the park managers and planners to consider the study's findings. The article describes the research process and structure explained well and repeatable in other parks. The study is original, and the research results can be transferred to other subtropical regions. The study includes fieldwork and observations of the study area. The paper is well-structured and well-written. The writing style corresponds to scientific writing. The text is understandable for researchers and the wider public, including park managers and planners. The conclusions present the study's findings and are provided in a structured order. Conclusions address the main research question and provide recommendations.
